# Gender Differences in Takotsubo Syndrome

**DOI:** 10.3390/biology11050653

**Published:** 2022-04-24

**Authors:** Tsutomu Murakami, Tomoyoshi Komiyama, Hiroyuki Kobayashi, Yuji Ikari

**Affiliations:** 1Department of Cardiology, School of Medicine, Tokay University, Isehara 259-1193, Japan; ikari@is.icc.u-tokai.ac.jp; 2Department of Clinical Pharmacology, School of Medicine, Tokay University, Isehara 259-1193, Japan; hkobayas@is.icc.u-tokai.ac.jp

**Keywords:** Takotsubo Syndrome, gender differences, emotional stress, physical stress

## Abstract

**Simple Summary:**

The manifestation of Takotsubo Syndrome (TTS) may be different in males and females based on past reports and our clinical research. However, the gender differences in TTS are unknown because patients with TTS are predominantly female. TTS is common in females; however, approximately 10–20% of males have TTS and it has been reported that in-hospital complications mostly occur in males. TTS in males is often caused by physical stress and often develops in the hospital or during hospitalization. TTS in males is associated with severe cardiac complications, which may require careful observations and interventions. Regarding the pathogenic mechanism of TTS, it has been reported that decreased estrogen levels, common in postmenopausal females, are involved in the pathogenic mechanism. Moreover, the pathological findings and gene expression were different in males and females. From these results, it can be considered that the mechanism of the onset of TTS may be different between males and females.

**Abstract:**

Most patients with Takotsubo Syndrome (TTS) are postmenopausal females. TTS in males is rare and gender differences have not been sufficiently investigated. Therefore, we investigated gender differences in TTS. TTS in males and females is often triggered by physical and emotional stress, respectively. Heart failure, a severe in-hospital complication, requires greater mechanical respiratory support in males. Fatal arrhythmias such as ventricular tachycardia and ventricular fibrillation and in-hospital mortality rates are higher in males. The white blood cell (WBC) count has been shown to be higher in males than in females with cardiovascular death compared with non-cardiovascular death. Therefore, the WBC count, a simple marker, may reflect severe TTS. Decreased estrogen levels, common in postmenopausal females, are a pathogenic mechanism of TTS. Females have a more significant increase in the extracellular matrix-receptor interaction than males. Moreover, the pathological findings after hematoxylin–eosin staining were different in males and females. Males had more severe complications than females in the acute phase of TTS; thus, more careful observations and interventions are likely required. From these results, it can be considered that the mechanism of the onset of TTS may be different between males and females. Therefore, it is necessary to fully understand the gender differences in order to more effectively manage TTS.

## 1. Introduction

Takotsubo Syndrome (TTS), a relatively rare disease, occurs in approximately 2–3% of patients with suspected acute coronary syndrome [1]. In addition, considering that the proportion of females is overwhelmingly high, the probability of encountering TTS in males is low, and their clinical features are unknown [2]. Reports on the gender differences in TTS are insufficient. Most worldwide analyses have used a large number of registered cases; however, only a small number of male patients with TTS were examined [3,4,5]. In this review, we investigated the gender differences in TTS. The mechanism of the onset of TTS is unclear and there is no specific treatment or prevention for TTS [6,7,8,9]. Therefore, we compared the similarities and differences between males and females with TTS and aimed to determine the reasons why the disease is overwhelmingly common in females, but more severe in males.

TTS was first discovered by a Japanese doctor in 1990 [10,11]. It was named Takotsubo because it is shaped like a takotsubo (octopus pot), a traditional trap for catching octopus in Japan [12]. The syndrome causes apical wall motion, which prevents a contraction during systole, resulting in the hypercontraction of the base of the heart. The disease is now known worldwide as Takotsubo cardiomyopathy or TTS [3,4,13,14,15,16,17]. The wall motion naturally improves in approximately two weeks after the onset with a good prognosis; therefore, there has been no progress in the research concerning the pathophysiology of TTS [18,19]. However, heart failure, acute mitral regurgitation, pulmonary hypertension, cardiogenic shock, and arrhythmia could lead to death in the acute phase from recent reports [20,21,22]. Furthermore, 3.7% of heart-related deaths occur during hospitalization and approximately 35% are due to heart failure [23]. However, the prognosis of a few hospitalized cases was not good [1,24]. We previously reported that TTS is more severe in males than in females [2]. This may be explained by the differences in stress responses between males and females because TTS is caused by stress. A decline in estrogen levels and the excess secretion of catecholamines may cause TTS, considering the high proportion of postmenopausal females with TTS [8,25]. TTS usually occurs in older postmenopausal women; therefore, decreased estrogen levels are thought to contribute to the onset of TTS and the theory remains promising [26,27]. However, the hypoestrogenic hypothesis cannot explain the mechanism of the onset of TTS in males. We previously reported that the rate of TTS associated with heart failure, arrhythmia, and in-hospital mortality is higher in males than in females [2]. Therefore, there may be another mechanism that increases the severity of TTS in males. Since the discovery of TTS, the detailed pathophysiology has not been elucidated and there is currently no cure or prevention for TTS. Therefore, a new approach is necessary. Hence, we observed the changes in the left ventricle myocardium during the acute phase via a left ventricle biopsy. A biopsy of the apical myocardium of the left ventricle during the acute phase of apical ballooning-type TTS was performed. The samples were analyzed using hematoxylin–eosin (H&E) staining and a DNA microarray [28].

We aimed to clarify the gender similarities and differences in TTS and to elucidate the disease onset mechanism. The clinical data including the clinical background, laboratory, echocardiography, and catheter data as well as the reported genetic analysis were obtained to determine the gender differences. The findings of this report will be very useful for the treatment of TTS in the acute phase and will contribute to the development of technologies for the accurate diagnosis of TTS. 

## 2. The Proportion of Genders and Diagnosis of TTS

As shown in Figure 1, the proportion of males with TTS varies from report to report; the number of males with TTS was small in each country. 

(1)As TTS is a relatively rare disease, most large-scale reports have used data from multicenter registries. Krishnamoorthy et al. [29] used the US National Inpatient Sample registry (20% registration of non-federal hospitals) [30] and Templin et al. used the registry data (The International Takotsubo Registry) of nine countries in Europe and the United States [4]. Our previous report [2] used the Tokyo Cardiovascular Care Unit (CCU) Network database (for those accommodated in CCUs at 71 facilities in Tokyo), and it is possible that severely ill patients were registered. Yoshizawa et al. [31] used the registry of 10 hospitals affiliated with 8 medical schools in eastern Japan that contained 10,622 cases with acute coronary syndrome (the Cardiovascular Research Consortium-8 Universities: CIRC-8U).

We selected the above four studies because they described the differences in TTS between males and females in detail. This is shown in Figure 1 below.

**Figure 1 biology-11-00653-f001:**
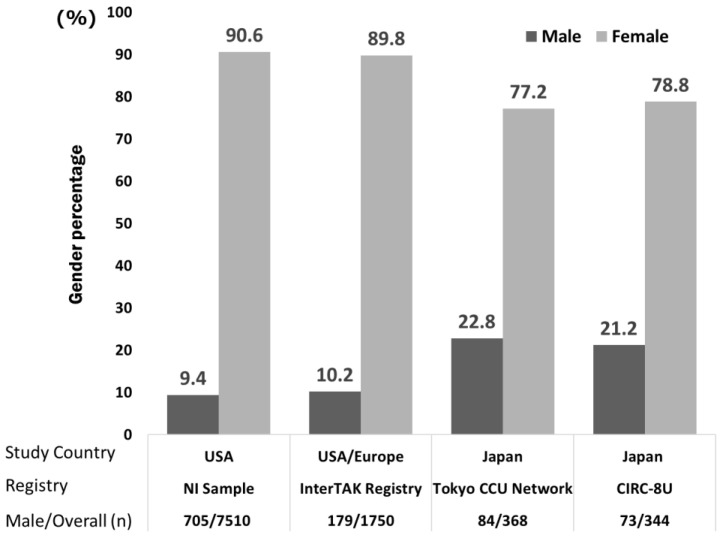
The gender differences in TTS in Japan, Europe, and the United States of America (USA). NI Sample, Nationwide Inpatient Sample; InterTAK Registry, The International Takotsubo Registry; CCU, Cardiovascular Care Unit; CIRC-8U, the Cardiovascular Research Consortium-8 Universities.

(2)From the 2008 modified Mayo Criteria [32], obstructive coronary artery disease is not an exclusion criterion for the diagnosis of TTS. Another major difference from the first edition of the Mayo Criteria is that after trauma, TTS can develop after an intracerebral hemorrhage (including a subarachnoid hemorrhage). However, it has been suggested that it may not be excluded and, therefore, not registered as TTS. In particular, male patients with coronary artery stenosis may be incorrectly diagnosed [13]. In addition, the subjective symptoms range from mild symptoms including chest discomfort to severe symptoms including respiratory distress from heart failure, shock, and left ventricular outflow tract stenosis. Mild cases might have been overlooked leading to the gender differences in TTS in each report. Currently, TTS is often diagnosed using the International Takotsubo Diagnostic Criteria (InterTAK Diagnostic Criteria) reported in 2018. Doctors have attempted to improve the accuracy of a TTS diagnosis using the InterTAK Diagnostic Score [33,34,35]. Recently, it has been reported that a differential diagnosis between TTS and acute myocardial infarction is possible due to the development of artificial intelligence in echocardiography [36]. Human monitoring is still necessary, but it is an area where further development is expected in the future [37].

## 3. Patient Backgrounds and Clinical Characteristics

### 3.1. Age

As shown in Table 1, the male patient group was younger than the female patient group. In addition, the average age of the patients from Japan was between 70 and 80 years whereas the patients from Europe and the United States were between 60 and 70 years of age, which could lead to an uneven ratio in the gender differences.

### 3.2. Preceding Stress and Symptoms at Admission

As shown in Table 2, there was a difference in preceding stress between the male and female patients. According to a report from the Tokyo CCU Network [2], 50.0% of males and 31.3% of females (*p* = 0.002) developed TTS due to physical stress and 19.0% of males and 31.0% of females (*p* = 0.039) developed the disease due to emotional stress. TTS in males and females was, therefore, caused by physical and emotional stress, respectively. Physical stress included acute respiratory failure, cerebrovascular accidents, infections, and postoperative trauma. Regarding the symptoms at admission, chest pain occurred in 39.3% of males and 51.4% of females (*p* = 0.051), but the difference was not significant. In addition, 35.7% of males and 32.8% of females had dyspnea (*p* = 0.613), but the difference was not significant. According to a report from the CIRC-8U [31], 64% of males and 46% of females (*p* = 0.007) developed the disease due to physical stress and 10% of males and 26% of females (*p* = 0.004) developed the disease due to emotional stress. Concerning the symptoms at admission, chest pain occurred in 42% of males and 50% of females (*p* = 0.292), but the difference was not significant. 

According to the reports from the International Takotsubo Registry [4], 50.8% of males and 34.3% of females (*p* < 0.001) developed the disease due to physical stress and 14.5% of males and 29.2% of females (*p* < 0.001) developed the disease due to emotional stress, which was similar to that reported in Japan. Physical stress included acute respiratory failure, postoperative fractures, cerebrovascular accidents, infections, cancer, and emotional stress including sadness, panic, fear, anxiety, interpersonal problems, anger, disappointment, financial difficulties, and unemployment. Concerning the symptoms at admission, chest pain occurred in 65.2% of males and 77.0% of females (*p* = 0.001), with a significant difference. In addition, 47.8% of males and 46.8% of females had dyspnea (*p* = 0.82), but the difference was not significant.

These reports confirmed that physical and emotional stress were the main causes of TTS in males and females, respectively. Chest pain and dyspnea were the most common symptoms and more females complained of chest pain than males.

### 3.3. Examination of the Blood Tests

At present, there are only a few blood tests for TTS that have been compared and examined by gender; the results are shown in Table 3, although the data are limited. 

The International Takotsubo Registry and CIRC-8U reported that the white blood cell (WBC) count (/μL) was higher in males than in females (10,680 (7650–15,600) vs. 9690 (7400–12,480); *p* = 0.013 [4] and 10,685 ± 4185 vs. 9704 ± 4853; *p* = 0.011 [31], respectively). The Tokyo CCU Network [2] reported no significant difference in the WBC count (/μL); however, the WBC tended to be higher in males than in females (9100 (7100–11,970) vs. 8100 (6400–11,000); *p* = 0.091).

Studies have reported that creatine kinase (CK) (a factor increase in the upper limit of the normal) at admission and peak CK (a factor increase in the upper limit of the normal) are higher in males, but the International Takotsubo Registry reported that there was no significant difference in the CK and peak CK values between males and females [4]. From the Tokyo CCU Network [2], the peak CK (IU/L) was higher in males than in females (471 (198–713) vs. 258 (143–394); *p* = 0.012); from the CIRC-8U [31], there was no significant difference in the peak CK (IU/L), but it tended to be higher in males (799 ± 1838 vs. 779 ± 2180; *p* = 0.065).

The International Takotsubo Registry and CIRC-8U reported that C-reactive protein (CRP) was higher in males than in females (5.00 (2.00–23.75) mg/L vs. 3.80 (1.13–11.00) mg/L; *p* = 0.021 [4] and 5.6 ± 7.1 vs. 2.7 ± 5.7; *p* < 0.001 [32], respectively). From the Tokyo CCU Network [2], there was no significant difference in CRP (mg/dL), but it tended to be higher in males than females (0.56 (0.1–3.0) vs. 0.32 (0.1–2.1); *p* = 0.055).

### 3.4. Examination of the Echocardiography

As shown in Table 4, most examples of TTS were the apical type and there was no difference between males and females. Regarding the left ventricular ejection fraction (LVEF) (%), the Tokyo CCU Network and CIRC-8U did not report any gender difference (48 (40–60) vs. 50 (40–64); *p* = 0.500 [2] and 44.7 ± 13.2 vs. 46.2 ± 13.0; *p* = 0.544 [31], respectively). The International Takotsubo Registry [4] reported a significant gender difference in cardiac dysfunction between males and females (39.0 ± 11.5 vs. 41.3 ± 11.8; *p* = 0.017), respectively. An apical thrombus was found in 2 of the 84 cases reported in the Tokyo CCU Network [2] and was more common in males.

### 3.5. The Co-Existence of Coronary Artery Disease

The Tokyo CCU Network [2] reported that 15.7% of males and 8.2% of females had coronary artery disease, but the difference was not significant (*p* = 0.061). The International Takotsubo Registry [4] reported that 21.1% of males and 14.7% of females had coronary artery disease and the difference was significant (*p* = 0.040). The CIRC-8U [31] reported that 19% of males and 5% of females had coronary artery disease and the difference was significant (*p* = 0.002).

### 3.6. Complications and Supportive Therapies during Hospitalization

Table 5 shows the details of the differences in complications and supportive therapies during hospitalization between male and female patients. According to the report from the Tokyo CCU Network, there was no significant difference in mortality between males and females (9.5% vs. 5.3%), but the mortality rate was higher in males [2]. The US National Inpatient Sample, International Takotsubo Registry, and CIRC-8U reported that the mortality rates were significantly higher in males than in females (4.8% vs. 2.1%; *p* = 0.04 [29], 7.3% vs. 3.8%; *p* = 0.025 [5], and 18% vs. 7%; *p* = 0.005 [31], respectively). The definition of heart failure varied from report to report, but heart failure was more common in males. In addition, cases requiring mechanical ventilation (invasive positive pressure ventilation or non-invasive positive pressure ventilation) were significantly higher in males in both the Tokyo CCU Network (28.6% vs. 12.7%; *p* < 0.05) [2] and the International Takotsubo Registry (29.5% vs. 16.0%; *p* < 0.001) [4]. Regarding the use of catecholamines, there was no difference between males and females reported by the Tokyo CCU Network (11.9% vs. 12.3%) [2], but the usage rate reported by the International Takotsubo Registry was significantly higher in males (21.0% vs. 11.2%; *p* < 0.001) [4]. Ventricular arrhythmias were more common in males, but the difference was not significant according to the US National Inpatient Sample (7.7% vs. 5.4%; *p* = 0.27) [29], Tokyo CCU Network (8.3% vs. 3.9%; *p* = not significant) [3], and CIRC-8U (5% vs. 4%; *p* = 0.510) [31]. In summary, males usually developed more severe complications and were more likely to require invasive interventions. 

### 3.7. Long-Term Outcomes and Therapies after Discharge 

According to the International Takotsubo Registry, the mortality rates per patient/year were significantly higher in males than in females (12.9% vs. 5.0%; *p* < 0.001) and there was no significant difference in TTS recurrence (0.8% vs. 1.9%; *p* = 0.22) [4]. In addition, the use of angiotensin-converting enzyme inhibitors or angiotensin II-receptor blockers at discharge was associated with an improved survival; survival was comparable between patients with and without beta-blockers from the retrospective registry data. In this paper, it should be noted that this analysis was from retrospective registry data based on discharge medication and has, therefore, inherent limitations. 

From a report that prospectively enrolled patients in the Takotsubo Italian Network, beta-blockers were associated with a significantly higher long-term survival within the median follow-up (24.0 months) [38]. However, there were no significant differences in terms of TTS recurrence and cardiac death. 

From the Danish nationwide registries, there was no significant difference between groups with respect to treatment with beta-blockers or angiotensin-converting enzyme inhibitors/angiotensin II-receptor blockers comparing the TTS recurrence of patients and those without TTS [39].

At present, there is no consensus on which drug is best for a long-term prognosis and the prevention of TTS recurrence. There is almost no mention of gender differences, so randomized controlled trial evidence is urgently needed [40]. 

It was reported that psychological anxiety was more common in patients with TTS and that the management thereof should be performed [40,41].

Moreover, it was reported that the long-term follow-up of patients with TTS revealed a rate of major adverse cardiac and cerebrovascular events of 9.9% per patient/year. The rate was higher in males than in females (16.0% vs. 8.7%; *p* = 0.002) [4]. Therefore, an aspirin treatment was considered to be effective. However, no association was found between aspirin use in TTS patients and a reduced risk of major adverse cardiac and cerebrovascular events at 30-day and 5-year follow-ups [42]. There is a supporting report that the presence of endothelial perturbation in females with TTS even at long-term from the index event by confirming the endothelial markers is involved in the vascular tone and in thrombosis as well as in residual platelet activation [43]. Patients with TTS must be monitored after discharge to reduce major adverse cardiac and cerebrovascular events and we believe that it is desirable to develop research on male as well as female patients.

## 4. Research Reports Supporting the Gender Differences in TTS

### 4.1. Effects of Estrogen Concentration

The cardioprotective effects of estrogen are well-known. Estrogen suppresses the RAS system and sympathetic nervous system. In an animal experiment using a TTS model, the group that received estrogen had a milder decrease in cardiac function and a milder increase in the pulse rate than the group that did not receive estrogen [26]. However, reports have suggested that estrogen contributes to the onset of TTS. In addition, in female humans matched by age and gender, the estrogen blood concentrations at the onset of TTS and myocardial infarctions were high [44]. Although a decrease in estrogen concentration contributes to the onset of TTS, it is possible that an increase at the time of onset may prevent the aggravation of TTS in females, but further studies on males are required.

### 4.2. Gene Expression, KEGG Analysis, and Pathological Analysis of the Heart Muscle

In an animal experiment of a model of myocarditis caused by the Coxsackie virus B3, there was a difference in the expression of the Toll-like receptors between males and females. It was reported that myocarditis was more severe in males even though the amount of the virus that spread to the heart muscle was the same [45]. Concerning the onset of the octopus trap syndrome, there was a difference in the response at the level of the myocardial gene expression and it was speculated that it may contribute to the difference in TTS between males and females.

We examined the differences in stress response in the myocardium between the genders in patients with TTS using a microarray analysis and a gene expression analysis based on KEGG [28]. For this study, the biopsied samples were obtained from over-70-year-old Japanese males and over-80-year-old Japanese females. Histopathological and DNA microarray analyses of the tissues from the left ventricle apex in the acute phase and apical ballooning-type were performed. KEGG analyses of the female patients confirmed the extracellular matrix (ECM)-receptor interaction pathway as well as the cell–cell interaction pathways such as cell adhesion molecules (CAMs) and the cytokine–cytokine-receptor interaction. The ECM-receptor and ECM-integrin interactions were strong (Table 6). This affected the interaction between the collagen, laminin, and integrin receptors. The KEGG analysis revealed that the ECM-receptor interaction was significantly increased in females than in males. We confirmed the changes in the expression levels of the various integrin subunits. 

We then performed a pathological analysis using the same tissues used during the microarray analysis, which were obtained from over-70-year-old males and over-80-year-old females. The biopsied specimens were fixed in a 10% neutral formalin solution overnight, dehydrated in an ethanol series, and cut into 6 μm-thick paraffin blocks using a microtome. The sections were dehydrated in distilled water and stained with H&E. 

The pathological findings after H&E staining were reported by Murakami et al. [28]. In the male patients with TTS, the necrosis area was broader with a lymphocytic infiltration and contraction band necrosis was observed with a high-power view. In contrast, in the female patients, only a few necrotic areas without a lymphocytic infiltration were observed.

## 5. Discussion

We investigated the gender differences in TTS using previous reports on TTS. Figure 1 shows a clear gender difference in TTS. More females had TTS in Japan, Europe, and the United States. A gender difference in TTS could be suggested just from these results.

The following factors could cause the gender difference in TTS. It has been shown that males often develop physical stress and females often develop emotional stress. In a recent report, it was stated that not only emotional stress and physical stress individually cause TTS, but a combination of both stresses also cause it [3]. Differentiating between the emotional and physical triggers of TTS would require a psychological evaluation to accurately recognize if the physical trigger did not also burden the emotional side of the patient [41]. Males who often develop TTS due to physical stress should be carefully monitored for psychological characteristics.

Although the reports of the studies vary, in-hospital death, heart failure, and life-threatening arrhythmias such as ventricular tachycardia (VT) or ventricular fibrillation (VF) were more common in males. Therefore, males are more likely to require cardiopulmonary supportive therapies such as respiratory support by mechanical ventilation or non-invasive positive pressure ventilation and catecholamine use. Recently, it was reported that Impella mechanical circulatory support was effective for TTS with a cardiogenic shock (systolic blood pressure was 90.1 ± 20.2 mmHg with the use of catecholamines) due to a low LVEF (19.4 ± 8.3%) until the recovery of the LVEF [46]. However, it may not be effective in TTS involving right ventricular failure [47] and reports of gender differences require further cases.

We reported that the group with a high WBC count was prone to in-hospital cardiac complications (cardiac death, VT, VF, thromboembolism, heart failure, and a severe atrioventricular block) and it was reported that these complications may indicate myocardial inflammation in TTS [23]. It may reflect inflammation caused by physical stress itself. However, the WBC count tended to be higher in patients with cardiovascular death than in those with non-cardiovascular death. Therefore, the WBC count, a simple conventional marker, could be used to identify severe TTS [23].

From the Tokyo CCU Network, a high WBC count and a male gender were the independent predictors of in-hospital composite cardiac events in TTS (cardiovascular death, severe pump failure defined using Killip III, and serious ventricular arrhythmias such as VT or VF) [2]. The International Takotsubo Registry and CIRC-8U reported that males had several in-hospital complications and high WBC counts.

One of the other causes of high mortality in males with TTS was physical stress. From the CIRC-8U [31], there was no significant difference in cardiovascular death (4% vs. 3%; *p* = 0.704), but there were differences in the other causes of death (14% vs. 4%; *p* = 0.003) between males and females. A study on the in-hospital outcomes by the preceding trigger [48] reported that there were no significant differences in cardiac death (2% vs. 2%; *p* = 1.00) between the physical and emotional stress triggers, but there were differences in all-cause death (11% vs. 2%; *p* < 0.001) between the physical and emotional stress triggers. Physical stress often occurs in males and the severity of the underlying disease is involved in the mortality rate.

It has been reported that 50% of non-cardiac deaths were complicated by heart failure; the presence of heart failure during the onset of TTS causes further aggravation and makes it difficult to create a better treatment plan [23]. Myocardial inflammation and heart failure may aggravate TTS differently for males and females.

Other studies have described the differences in the expression of the Toll-like receptors between males and females [45]. In addition, microarray and gene expression analyses of left ventricular biopsy samples in males and females were performed. A KEGG analysis revealed that the ECM-receptor interaction in females significantly increased compared with males. We confirmed the changes in the expression levels of the various integrin subunits [28].

Moreover, the pathological findings after H&E staining of the left ventricular biopsy samples were different between males and females [28]. In males with TTS, the area of necrosis was broader with a lymphocytic infiltration and contraction band necrosis was found with a high-power view. In contrast, only a few necrotic areas without a lymphocytic infiltration were observed in females. 

There are an increasing number of reports of an association between TTS and COVID-19. TTS in COVID-19 can be primary TTS due to the emotional stress associated with the global pandemic or secondary TTS due to a COVID-19 infection [49]. Patients who were hospitalized for COVID-19 pneumonia and subsequently developed TTS due to physical stress have been reported to have a poor prognosis [50]. In this report, there were 7 patients (4 males and 3 females) with TTS due to a COVID-19 infection. A total of 6 patients (4 males and 2 females) required respiratory support by intubation and 4 patients (2 males and 2 females) died. As the mortality of patients with secondary TTS due to COVID-19 is high, it is important to ensure an early diagnosis for COVID-19-infected patients and we believe that further cases need to be examined in the future.

## 6. Conclusions

Possible differences may exist between males and females based on past reports and our clinical research. The review of our research contributes to new developments in future studies. It should be noted that TTS in males becomes severe. We aim to elucidate the mechanisms of the development of TTS using a larger sample size to increase the reliability of the data in the future.

## Figures and Tables

**Table 1 biology-11-00653-t001:** The differences in age between males and females.

Country	Registry	Study Period	All, Years	Male, Years	Female, Years	*p*-Value	Ref.
USA	NI Sample	2009–2010	65.6 (64.9–66.2)	59.5 (56.6–62.3)	66.2 (65.5–66.8)	<0.001	[29]
USA/Europe	InterTAK Registry	1998–2014	66.4 ± 13.1	62.9 ± 13.1	66.8 ± 13.0	<0.001	[4]
Japan	Tokyo CCU Network	2010–2012	76 (67–82)	72 (64–81)	76 (68–83)	0.040	[2]
Japan	CIRC-8U	1997–2014	71.6 ± 11.2	71.8 ± 10.4	71.5 ± 11.4	0.899	[31]

USA, United States of America; NI Sample, Nationwide Inpatient Sample; InterTAK Registry, The International Takotsubo Registry; CCU, Cardiovascular Care Unit; CIRC-8U, the Cardiovascular Research Consortium-8 Universities; Ref., reference.

**Table 2 biology-11-00653-t002:** The differences in preceding stress between the male and female patients.

Registry		Male	Female	*p*-Value	Ref.
InterTAK Registry	Physical stress, %	50.8	34.3	<0.001	[4]
	Emotional stress, %	14.5	29.2	<0.001	
	Absence of stress, %	25.7	28.8	0.39	
Tokyo CCU Network	Physical stress, %	50.0	31.3	0.002	[2]
	Emotional stress, %	19.0	31.0	0.039	
	Absence of stress, %	31.0	37.7	0.260	
CIRC-8U	Physical stress, %	64	46	0.007	[31]
	Emotional stress, %	10	26	0.004	
	Absence of stress, %	26	28	0.764	

InterTAK Registry, The International Takotsubo Registry; CCU, Cardiovascular Care Unit; CIRC-8U, the Cardiovascular Research Consortium-8 Universities; Ref., reference.

**Table 3 biology-11-00653-t003:** The differences in examinations of blood tests between male and female patients.

Registry		Male	Female	*p*-Value	Ref.
InterTAK Registry	WBC (/μL)	10,680 (7650–15,600)	9690 (7400–12,480)	0.013	[4]
	CRP (mg/L)	5.00 (2.00–23.75)	3.80 (1.13–11.00)	0.021	
Tokyo CCU Network	WBC (/μL)	9100 (7100–11,970)	8100 (6400–11,000)	0.091	[2]
	Peak CK (IU/L)	471 (198–713)	258 (143–394)	0.012	
	BNP (pg/mL)	233 (75–521)	199 (76–627)	0.855	
	CRP (mg/dL)	0.56 (0.1–3.0)	0.32 (0.1–2.1)	0.055	
CIRC-8U	WBC (/μL)	10,685 ± 4185	9704 ± 4853	0.011	[31]
	Peak CK (IU/L)	799 ± 1838	779 ± 2180	0.065	
	CRP (mg/dL)	5.6 ± 7.1	2.7 ± 5.7	<0.001	

InterTAK Registry, The International Takotsubo Registry; CCU, Cardiovascular Care Unit; CIRC-8U, the Cardiovascular Research Consortium-8 Universities; WBC, white blood cell count; CK, creatinine kinase; BNP, brain natriuretic peptide; CRP, C-reactive protein; Ref., reference.

**Table 4 biology-11-00653-t004:** The differences in echocardiography images between the male and female patients.

Registry		Male	Female	*p*-Value	Ref.
InterTAK Registry	Apical type, %	81.6	81.7	0.96	[4]
	Midventricular type, %	12.8	14.8	0.49	
	LVEF (%)	39.0 ± 11.5	41.3 ± 11.8	0.017	
Tokyo CCU Network	Apical type, %	90.5	90.8	0.918	[2]
	Midventricular type, %	N/A	N/A	N/A	
	LVEF (%)	48 (40–60)	50 (40–64)	0.500	
	LVOTO, %	4.8	9.2	0.196	
CIRC-8U	Apical type *, %	93.6	91.0	NS	[31]
	Midventricular type *, %	2.1	4.0	NS	
	LVEF (%)	44.7 ± 13.2	46.2 ± 13.0	0.544	
	LVOTO *, %	0	6	0.162	

InterTAK Registry, The International Takotsubo Registry; CCU, Cardiovascular Care Unit; CIRC-8U, the Cardiovascular Research Consortium-8 Universities; LVEF, left ventricular ejection fraction; LVOTO, left ventricular outflow tract obstruction; * these factors were diagnosed by cardiac catheterization; N/A, not available; NS, not significant; Ref., reference.

**Table 5 biology-11-00653-t005:** The differences in complications and supportive therapies during hospitalization between male and female patients.

Registry		Male	Female	*p*-Value	Ref.
NI Sample	Mortality, %	4.8	2.1	0.04	[29]
	Respiratory failure, %	18.2	12.6	0.06	
	Ventricular arrhythmias, %	7.7	5.4	0.27	
InterTAK Registry	Mortality, %	7.3	3.8	0.025	[4]
	Respiratory support, %	29.5	16.0	<0.001	
	Catecholamine use, %	21.0	11.2	<0.001	
Tokyo CCU Network	Mortality, %	9.5	5.3	NS	[2]
	Heart failure *, %	20.2	10.6	<0.05	
	Ventricular arrhythmias, %	8.3	3.9	NS	
	Respiratory support, %	28.6	12.7	<0.05	
	Catecholamine use, %	11.9	12.3	NS	
CIRC-8U	Mortality, %	18	7	0.005	[31]
	Cardiovascular death, %	4	3	0.704	
	Death by other reasons, %	14	4	0.003	
	Heart failure, %	34	29	0.388	
	Ventricular arrhythmias, %	5	4	0.510	

NI Sample, Nationwide Inpatient Sample; InterTAK Registry, CCU, Cardiovascular Care Unit; The International Takotsubo Registry; CIRC-8U, the Cardiovascular Research Consortium-8 Universities; NS, not significant; *, heart failure was defined as Killip ≥ Ⅲ; Ref., reference.

**Table 6 biology-11-00653-t006:** The results of the Kyoto Encyclopedia of Genes and Genomes analysis of female patients.

Category	Term	Count	%	*p*-Value	Bonferroni	Benjamini
KEGG_PATHWAY	hsa04512: ECM-receptor interaction	17	2.007084	1.23 × 10^−7^	2.96 × 10^−5^	2.96 × 10^−5^
KEGG_PATHWAY	hsa04514: Cell adhesion molecules (CAMs)	17	2.007084	8.61 × 10^−5^	0.020544	0.010326
KEGG_PATHWAY	hsa04060: Cytokine–cytokine-receptor interaction	19	2.243211	0.004895	0.693482	0.325753

## Data Availability

The data presented in this study are available in the article.

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
