# Peer review of "Gender Differences in Takotsubo Syndrome"

_biology, 2022, doi:10.3390/biology11050653_

Round 1

Reviewer 1 Report

The review entitled “Gender differences in Takotsubo syndrome” discussed the mechanisms underlying gender differences in Takotsubo Syndrome including pathological findings and gene expression. Below are comments that will further strengthen the review:

  1. There is a lack of discussion on evidence-based interventions as well as potential clinical treatment and therapeutics. Please refer to the Dawson update on Circulation (PMID: 35344411), beta-blockers (PMID:35361673) and percutaneous left ventricular assist device (PMID:35365425). More specifically, any new discoveries in gender different treatment and management in the field?
  1. New discoveries on Takotsubo and Covid 19 is exciting and worth mentioning in the discussion. (e.g. PMID: 33759445)
  2. The authors should also mention briefly the new technologies involved in diagnosis: Deep learning (PMID:35353132) and AI (PMID:35353118).

Author Response

In Reply to Editors

We deeply appreciate the Editor and Reviewers for their thoughtful comments, which have provided us with valuable opportunities to improve our work. We have carefully considered the comments given by the Reviewers and have addressed them point-by-point below. The changes in the revised manuscript are highlighted in yellow.

Reply to Reviewer 1

We deeply appreciate your time and input. We have carefully considered your comments and addressed them.

The review entitled “Gender differences in Takotsubo syndrome” discussed the mechanisms underlying gender differences in Takotsubo Syndrome including pathological findings and gene expression. Below are comments that will further strengthen the review:

  1. There is a lack of discussion on evidence-based interventions as well as potential clinical treatment and therapeutics. Please refer to the Dawson update on Circulation (PMID: 35344411), beta-blockers (PMID:35361673) and percutaneous left ventricular assist device (PMID:35365425). More specifically, any new discoveries in gender different treatment and management in the field?

Reply:

Author’s response: We appreciate your comment.

We have added a sentence about Circulation in the new section (3,7 Long term outcomes and therapies after discharge) on page 8 lines 286-290 as follows:

At present, there is no consensus on which drug is better for long-term prognosis and prevention of TTS recurrence, and there is almost no mention of gender differences, so randomized controlled trial evidence is urgently needed [PMID: 35344411]. In addition, it was reported that psychological anxiety was more common in patients with TTS and that management thereof should be performed [PMID: 35344411].

We have added a sentence about beta-blockers in the new section (3,7 Long term outcomes and therapies after discharge) on page 8 lines 278-281 as follows:

From a report which prospectively enrolled patients in the Takotsubo Italian Network, beta-blockers were associated with a significantly higher long-term survival within the median follow-up (24.0 months) [PMID:35361673]. However, there were no significant differences in terms of TTS recurrence and cardiac death.

We have added a sentence about percutaneous left ventricular assist device in the Discussion section on page 10, lines 363-368 as follows:

Recently, it was reported that Impella mechanical circulatory support was effective for TTS with cardiogenic shock (systolic blood pressure was 90.1 ± 20.2 mmHg with use of catecholamines) due to low LVEF (19.4 ± 8.3 %) until recovery of LVEF [PMID:35365425]. However, it may not be effective in TTS involving the right ventricular failure, and reports of gender differences require further cases.

Unfortunately, there are no new discoveries in gender different therapies for the acute phase and chronic phase. At present, there is no consensus on which drug is better for long-term prognosis and prevention of recurrence, and there is almost no mention of gender differences.

  1. New discoveries on Takotsubo and Covid 19 is exciting and worth mentioning in the discussion. (e.g. PMID: 33759445)

Reply:

Author’s response:) We appreciate your comment. We have added a sentence in the Discussion section on page 11, lines 405-414 as follows:

Recently, there are increasing reports of an association between TTS and COVID-19. TTS in COVID-19 can be primary TTS due to the emotional stress associated with the global pandemic, or secondary TTS due to COVID-19 infection [PMID: 33759445]. Especially, patients who were hospitalized for COVID-19 pneumonia and subsequently developed TTS due to physical stress have been reported to have a poor prognosis. In this report, there were seven patients (4 males and 3 females) with TTS due to COVID-19 infection. Six patients (4 males and 2 females) required respiratory support by intubation and 4 patients (2 males and 2 females) died. Since mortality of patients with secondary TTS due to COVID-19 is high, it is important to ensure an early diagnosis for COVID-19 infected patients and we believe that further cases need to be examined in the future.

  1. The authors should also mention briefly the new technologies involved in diagnosis: Deep learning (PMID:35353132) and AI (PMID:35353118).

Reply:

Author’s response: We appreciate your comment. We have added a sentence in the proportion of the gender and diagnosis of TTS on page 3, line 128 – page 4, line131, as follows:

Recently, it has been reported that the differential diagnosis between TTS and acute myocardial infarction is possible due to the development of artificial intelligence in echocardiography [PMID:35353118]. Human monitoring is still necessary, but it is an area where further development is expected in the future [PMID:35353132]

Reviewer 2 Report

The manuscript entitled "gender differences in Takotsubo syndrome"  highlights potential clinical differences in presentation.

the authors refer to the decrease in estrogen level as one of the most relevant causes of TS onset in postmenopausal women. I believe the authors: 

1. should address physiopathologic mechanisms and observations published by Amadio et al (Amadio P, et al. Persistent long-term platelet activation and endothelial perturbation in women with Takotsubo syndrome. Biomed Pharmacother. 2021 Apr;136:111259. doi: 10.1016/j.biopha.2021.111259) displaying alterations in the production of
endothelial markers involved in vascular tone and thrombosis, as well
as residual platelet activation, despite low dose ASA treatment.
T Indeed, previous studies show that 3 months after the
acute event, TTS patients show an activated platelet response, which
is more marked in those with the highest levels of plasma
catecholamines. The study provided evidence that, even in the long-term, the residual platelet thromboxane biosynthesis, is significantly greater than that of women with CAD, thus suggesting the occurrence in TTS women of a reduced capacity of ASA to fully inhibit platelet thromboxane formation

2. consider paper from Butt JH, Bang LE, Rørth R, Schou M, Kristensen SL, Yafasova A, Havers-Borgersen E, Vinding NE, Jessen N, Kragholm K, Torp-Pedersen C, Køber L, Fosbøl EL. Long-term Risk of Death and Hospitalization in Patients With Heart Failure and Takotsubo Syndrome: Insights From a Nationwide Cohort. J Card Fail. 2022 Feb 12:S1071-9164(22)00048-3. , evaluating long-term outcomes and mortality risk post-TTS

3. the more frequent emotional stressor in women as TTS causes should be addressed in the discussion section, according to a recent analysis from Gorini et al Psychological Characteristics of Patients with Takotsubo Syndrome and Patients with Acute Coronary Syndrome: An Explorative Study toward a Better Personalized Care. J Pers Med. 2022 Jan 4;12(1):38. doi: 10.3390/jpm12010038. 

Author Response

In Reply to Editors

We deeply appreciate the Editor and Reviewers for their thoughtful comments, which have provided us with valuable opportunities to improve our work. We have carefully considered the comments given by the Reviewers and have addressed them point-by-point below. The changes in the revised manuscript are highlighted in yellow.

Reply to Reviewer 2

We deeply appreciate your time and input. We have carefully considered your comments and addressed them.

  1. should address physiopathologic mechanisms and observations published by Amadio et al (Amadio P, et al. Persistent long-term platelet activation and endothelial perturbation in women with Takotsubo syndrome. Biomed Pharmacother. 2021 Apr;136:111259. doi: 10.1016/j.biopha.2021.111259) displaying alterations in the production of endothelial markers involved in vascular tone and thrombosis, as well as residual platelet activation, despite low dose ASA treatment. T Indeed, previous studies show that 3 months after the acute event, TTS patients show an activated platelet response, which is more marked in those with the highest levels of plasma catecholamines. The study provided evidence that, even in the long-term, the residual platelet thromboxane biosynthesis, is significantly greater than that of women with CAD, thus suggesting the occurrence in TTS women of a reduced capacity of ASA to fully inhibit platelet thromboxane formation

Reply:

Author’s response: We appreciate your comment. We have added a sentence in the new section (3,7 Long term outcomes and therapies after discharge) on page 8 lines 291-300 as follows:

Moreover, it was reported that long-term follow-up of patients with TTS revealed a rate of major adverse cardiac and cerebrovascular events of 9.9% per patient-year [5]. Therefore, aspirin treatment was considered to be effective. However, no association was found between aspirin use in TTS patients and a reduced risk of major adverse cardiac and cerebrovascular events at 30-day and 5-year follow-up. There is a supporting report that the presence of endothelial perturbation in females with TTS even at long-term from the index event by confirming endothelial markers involved in vascular tone and in thrombosis, as well as residual platelet activation [Biomed Pharmacother. 2021 Apr;136:111259]. Patients with TTS must be monitored after discharge to reduce major adverse cardiac and cerebrovascular events and we believe that it is desirable to develop research on male as well as female patients.

  1. consider paper from Butt JH, Bang LE, Rørth R, Schou M, Kristensen SL, Yafasova A, Havers-Borgersen E, Vinding NE, Jessen N, Kragholm K, Torp-Pedersen C, Køber L, Fosbøl EL. Long-term Risk of Death and Hospitalization in Patients With Heart Failure and Takotsubo Syndrome: Insights From a Nationwide Cohort. J Card Fail. 2022 Feb 12:S1071-9164(22)00048-3. , evaluating long-term outcomes and mortality risk post-TTS

Reply:

Author’s response: We appreciate your comment. We have added a sentence in the new section (3,7 Long term outcomes and therapies after discharge) on page 8, lines 282-285 as follows:

From the Danish nationwide registries, there was no difference between groups with respect to treatment with beta-blockers or angiotensin-converting enzyme inhibitors/angiotensin II receptor blockers comparing TTS recurrence patients and those without [J Card Fail. 2022 Feb 12:S1071-9164(22)00048-3.].

  1. the more frequent emotional stressor in women as TTS causes should be addressed in the discussion section, according to a recent analysis from Gorini et al Psychological Characteristics of Patients with Takotsubo Syndrome and Patients with Acute Coronary Syndrome: An Explorative Study toward a Better Personalized Care. J Pers Med. 2022 Jan 4;12(1):38. doi: 10.3390/jpm12010038.

Reply:

Author’s response: We appreciate your comment. We have added a sentence in the Discussion section on page 10, lines 352-358 as follows:

In a recent report, it was stated that not only emotional stress and physical stress alone can cause TTS, but a combination of both stresses also cause it. Also differentiating between emotional and physical triggers of TTS would require a psychological evaluation to accurately recognize if the physical trigger did not also burden the emotional side of the patient [J Pers Med. 2022 Jan 4;12(1):38]. Males who often develop TTS due to physical stress should be carefully monitored for psychological characteristics.
